# Discretization of the Urban and Non-Urban Shape: Unsupervised Machine Learning Techniques for Territorial Planning

Lorena Fiorini *, Federico Falasca, Alessandro Marucci and Lucia Saganeiti

Department of Civil, Construction-Architecture and Environmental Engineering, University of L'Aquila, Via G. Gronchi, 18, 67100 L'Aquila, Italy
* Correspondence: lorena.fiorini@univaq.it

**Abstract:** One of the goals of the scientific community is to equip the discipline of spatial planning with efficient tools to handle huge amounts of data. In this sense, unsupervised machine learning techniques (UMLT) can help overcome this obstacle to further the study of spatial dynamics. New machine-learning-based technologies make it possible to simulate the development of urban spatial dynamics and how they may interact with ecosystem services provided by nature. Modeling information derived from various land cover datasets, satellite earth observation and open resources such as Volunteered Geographic Information (VGI) represent a key structural step for geospatial support for land use planning. Sustainability is certainly one of the paradigms on which planning and the study of past, present and future spatial dynamics must be based. Topics such as Urban Ecosystem Services have assumed such importance that they have become a prerogative on which to guide the administration in the difficult process of transformation, taking place not only in the urban context, but also in the peri-urban one. In this paper, we present an approach aimed at analyzing the performance of clustering methods to define a standardized system for spatial planning analysis and the study of associated dynamics. The methodology built ad hoc in this research was tested in the spatial context of the city of L'Aquila (Abruzzo, Italy) to identify the urbanized and non-urbanized area with a standardized and automatic method.

**Keywords:** unsupervised machine learning; UMLT; urban planning; clustering models; machine learning; mosaic urban pattern

## 1. Introduction

New planning uses innovative and technologically advanced tools through the continuous flow of data and simulations of possible scenarios. In this application context, approaches based on GIS technologies and methodologies play a fundamental role, both in vulnerability assessment and in supporting decision-making processes as a whole and in the analysis of risk, impact and consequences [1,2]. In the last years, the scientific community has focused increasingly on the application implications of the use of GIS-based technologies, and the need for effective spatial standards and interfaces, spatial analysis tools and integrated hardware/software platforms (Spatial Data Infrastructure, SDI) have played a key role in the development of these activities [3–5].

GIS-based models have been used to quantify both the demand and provision of Urban Ecosystem Services (UES) [6–8]. These models allow, through spatial data such as land cover and land use (LULC), for the quantification of UESs by associating and comparing them with vegetation types and other landscape features [9]. Moreover, spatial dynamics can reveal heterogeneity and trends in the distribution of UESs over urban landscapes, which can be of importance for urban sustainability planning [10]. Other studies have quantified spatial variation in UES values using a hedonic price model and have analyzed spatial relations among biodiversity features to assess habitat supply [11].

The GIS-based approach to spatial planning allows one to benefit from ICT technologies. In particular, the new research activities in this area concern the application of indicators, engineering techniques aimed at identifying appropriate indices able to return information on the reaction of urban tissues with respect to environmental disturbances that determine impact/risk scenarios [12,13]. The infrastructures that today aim to collect and manage the immense amount of information produced by different systems express their real power in integration and interoperability [5,14]. New applications and innovative geospatial services are the basis of this research, with reference to spatial analysis and fast monitoring tools for urban resilience, the redevelopment of sensitive areas to combat hydraulic and hydrogeological risk; for the mitigation of climate change; for the improvement of/increase in ecosystem services; and for the evaluation of the carbon balance. All these issues are at the basis of the correct application of legislation on the Strategic Environmental Assessment (SEA), which should normally regulate the effects of urban planning [15,16]. The same procedure is based on the formulation of alternative assessment scenarios and on the concepts of mitigation and compensation [17]. Therefore, through a technological and cultural reform, it is possible to make SEA an effective and fundamental tool for urban planning. In this particularly active context, the experimentation of technologies in territorial sciences assumes a fundamental importance for the development of the field of territorial planning.

Indeed, today, Machine learning (ML) has increasingly been used in studies concerning territorial sciences and urban dynamics [18–21]. Neural networks were introduced in the 1950s, but only recently they have reached advanced processing power and data storage capabilities to the point that deep learning (DL) algorithms can be used to create new applications, including satellite imagery analysis [22]. Therefore, experimentation with ML and DL can only increase the interpretive capabilities of the urban mosaic and can generate sustainable territorial configurations [23,24].

This work exploits unsupervised machine learning techniques to discriminate urbanized from non-urbanized areas, a key step in implementing an urban and peri-urban mosaic pattern evaluation system.

The innovation of this research is using a shift of the hexagonal grid in order to bypass the randomized approach of a static grid and to evaluate which result can best approximate the structure of the urbanized space. The goal is to define the area of highest probability of intercepting the urbanized tiles through image analysis.

This paper is part of a broader work that aims at defining ecosystem functions inside and outside urban areas and thus at the design and provision of related ecosystem services.

## 2. Materials and Methods

### 2.1. Study Area and Dataset

The study area is represented by the city of L'Aquila, situated in the Abruzzo region (central Italy) (Figure 1).

This territorial scope includes both the natural system composed by protected areas (PAs), established at the local and national/international level (Natura 2000 network, local protected areas, Important Birding Areas, Ramsar areas, etc.), and by the anthropic system, consisting mostly of artificial areas. However, the latter do not have net defined boundaries due to the territorial morphology that not only influences aspects of the planning action, such as the dynamics of urban development and the protection of the natural matrix, but also contributes to mixing up the functionalities and the intrinsic characteristics of the two systems, resulting in a more complex pattern [25].

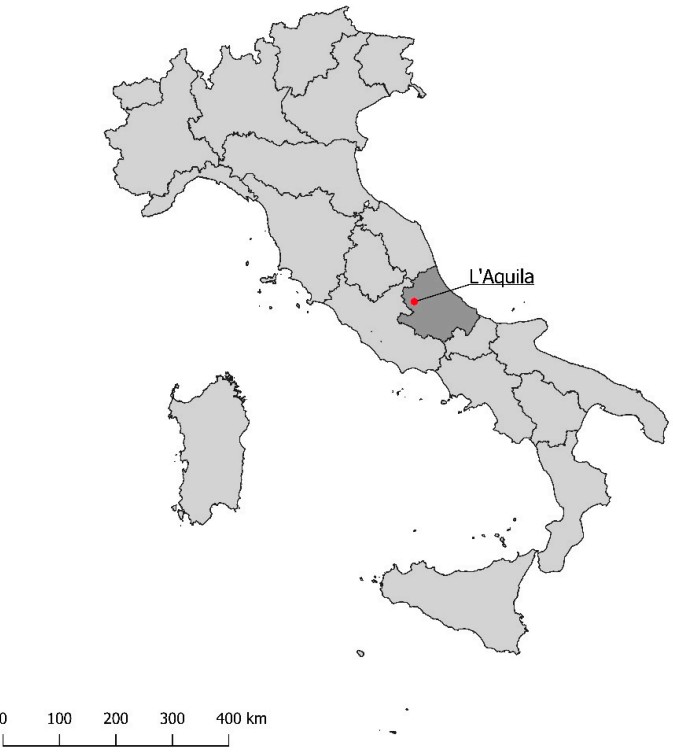

**Figure 1.** Study area.

In this peculiar context, the sustainable development principle entails a fundamental prerogative for the planning policies, to the point that several projects and plans are envisaged for this and other neighboring territorial scopes. LIFE IMAGINE is an integrated LIFE project aimed at sustaining the development of an integrated and unified strategy for the management of the Natura 2000 network in the Umbria region through actions to foster the connection, protection and enhancement of agricultural lands, forest habitats and wetlands, and the conservation of severe animal species [26]. Similarly, the "Sustainability, resilience, adaptation for the protection of the ecosystems and the physical reconstruction in central Italy" (SostEn&Re) project that concludes in autumn 2022 represents an important experience for the Abruzzo, Marche and Umbria regions in the formulation of guidelines to promote the implementation of the National Strategy for sustainable development [27]. The accomplishment of these projects brings the planning policies to a new level of concern about the aspects of sustainable development, creating new tools and supporting strategies to help local administration implement sustainable development goals set by both the national and international levels [28].

Despite this, on the other hand, municipalities manage territorial transformations with planning tools drafted, in certain cases, almost fifty years ago, defining threats and/or pressure on the network formed by all the possible connections between PAs. The city of L'Aquila does not escape these dynamics, which results in even more complexity if considered in a territorial scope, for which different thrusts of development subsist. Indeed, to put in place solutions aimed at enhancing the best planning practices, different projects and tools set both at the national and EU levels are envisaged; e.g., NextGenerationEU established two important instruments with the aim of involving member states in significant changes both from an environmental and a social point of view (Recovery and Resilience Facilities and Recovery Assistance for Cohesion and the Territories of Europe) [29]. These two instruments, aimed at the sustainable development of European Union member countries and the revitalization of their own economy, were followed up in Italy by the National Recovery and Resilience Plan (PNRR), which, in its current state, stimulates a series of interventions for the development of the country [30].

In the depicted framework, studies such as the current one, aimed at distinguishing urbanized cells from the natural ones, play a fundamental role as tools to address and formulate efficient planning policies.

The dataset utilized to perform the analyses is the Urban Atlas (UA), a product from the Copernicus Land Monitoring Service, in which the categories represent different Functional Urban Areas (FUA). The year of the utilized UA is 2018, and the spatial resolution is of 0.25 hectares for the first class and of 1 hectare for the other ones (class 2–5), with a minimum mapping width of 10 m [31]. The European Urban Atlas provides land use patterns of major European cities, so it is well suited to perform a comparative analysis. Being used in different kinds of research, these data manage also reveal/highlight important connections such as human activities and land use [32–34]. For example, UAs were also used to compare patch perimeter metrics of Lisbon (Portugal), Barcelona (Spain), Rome (Italy) and Athens (Greece), four large metropolitan regions in southern Europe.

### 2.2. Grid Generation Procedures

In the present work, a hexagonal mesh grid (cells) was used to discretize the study area. Currently, rasters in a GIS-based environment often follow a rectangular-shaped grid. Hence, remotely sensed images should be rectified in hexagonal grids when the objective/aim is to return a better representation of the neighborhood relations (Figure 2) [35,36]. Compared to the rectangular ones, hexagonal grid cells align along three axes, also having a more variable shape [37]. Due to the "meet points" set at their edges, hexagonal grid cells are also less ambiguous. Indeed, rectangular grid lines merge their continuity to the orthogonal neighbor cells, and, being sensitive to lines parallel to the x or y axes, human vision becomes distracted by them [38].

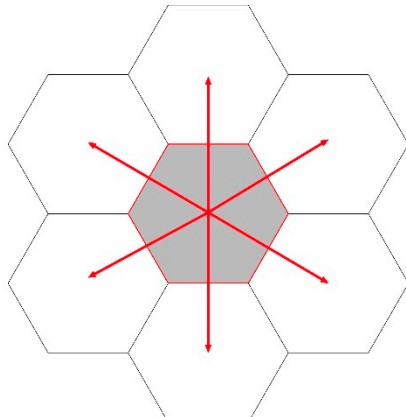

**Figure 2.** Axis alignment for a hexagonal-shaped grid.

Hence, this work fits into the current of Cellular Automata (CA) model approaches, performed through hexagonal-shaped grids, a type of geometry already used in various studies such as the current one [39–41].

First, we created a hexagonal-shapde grid in a GIS-based environment (QGIS software, version 3.16.3, Hannover, Germany) [42], with the extension of the layer represented by the UA. The cell size was set according to the peri-urban areas' perception capacity. The spatial and structural definitions of such areas are still an open discussion for spatial sciences; therefore, the area of the single hexagon was set to 2 square km, arbitrarily set based on doubling the mean urban area values detectable in the 2018 Corine Land Cover dataset [31].

Grid creation in a GIS-based environment is set on the UA extension, so it is not optimized for the best discretization of urban and nonurban areas. To evaluate the best grid position for our analysis, we applied a 3-directional shift, resumable with the following abbreviations (Figure 3):

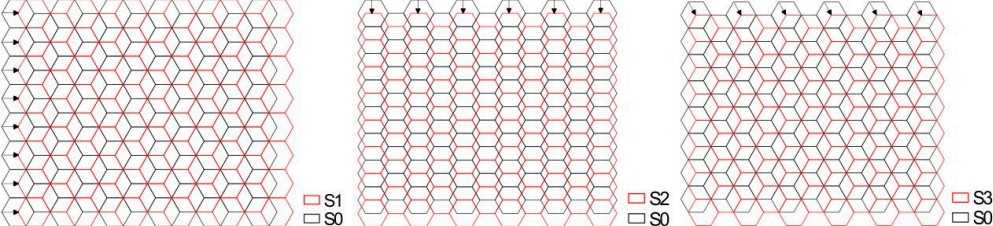

**Figure 3.** Grid Shifts. $S_1$: Right shift, equal to the radius of the circumscribed circumference of the single hexagonal cell; $S_2$: Down shift, equal to the apothem of the hexagonal cell; $S_3$: Diagonal shift, equal to the radius of the circumscribed circumference of a single hexagonal cell.

$S_0$—Null shift.

$S_1$—Right shift, equal to the radius of the circumscribed circumference of the single hexagonal cell.

$S_2$—Down shift, equal to the apothem of the hexagonal cell.

$S_3$—Diagonal shift, equal to the radius of the circumscribed circumference of a single hexagonal cell.

The reason that it was chosen to apply to only these 3 shifts is the redundancy of the translation. In fact, additional movements made with the specified distances lead to one of the four initial grid conditions.

The Urban Atlas was reorganized into two datasets for data processing. First, we considered the division between artificial areas (class 1, 17 urban sub-classes) and non-artificial areas (classes 2–5, 10 non-urban sub-classes). Then, we performed the same analyses with the five categories of the UA (Table 1).

**Table 1.** Urban Atlas (UA) class distinction.

| 2 UA CLASSES | 5 UA CLASSES |
|---|---|
| 1 Artificial areas | 1 Artificial areas |
| 2 Non-artificial areas | 2 Agricultural areas<br>3 Natural areas<br>4 Wetlands<br>5 Water bodies |

After the intersection between the UA and every grid ($S_0$, $S_1$, $S_2$, $S_3$) (Figure 4), the areas occupied by individual categories in each hexagon were aggregated. It is important to notice that, for the dataset, only cells with a 2 square km area were used to avoid all the mosaic tiles that, due to the intersection between the different grids and the UA, could not be considered like the larger ones. This operation resulted in 606 tiles for the $S_0$ shift; 605 tiles for the $S_1$ shift; 598 tiles for the $S_2$ shift; and 599 tiles for the $S_3$ shift.

Finally, the process from which the grids were obtained was standardized using the integrated graphical modeler option in QGIS. No specific plugins were used.

### 2.3. Unsupervised Machine Learning Procedures

Once all the necessary data were obtained, we performed unsupervised machine learning (UML) analyses to separate the urbanized cells from the natural ones.

The software used for this purpose was GeoDa, an open-source and free program, developed for those approaching spatial analysis. GeoDa allows for the deepening and exploration of spatial models and distributions [43]. Particularly, the models used to perform the analyses of interest were those well deepened/established in the scientific literature, which belong to the C clustering library and which GeoDa refers to as the following [44]: K—Means; K—Medians; K—Medoids (CLARA Algorithm); K—Medoids; (PAM Algorithm); Spectral K-NN; Spectral Mutual K-NN; Spectral Gaussian; Hierarchical Single linkage; Hierarchical Ward's linkage; Hierarchical Complete linkage; and Hierarchical Average linkage.

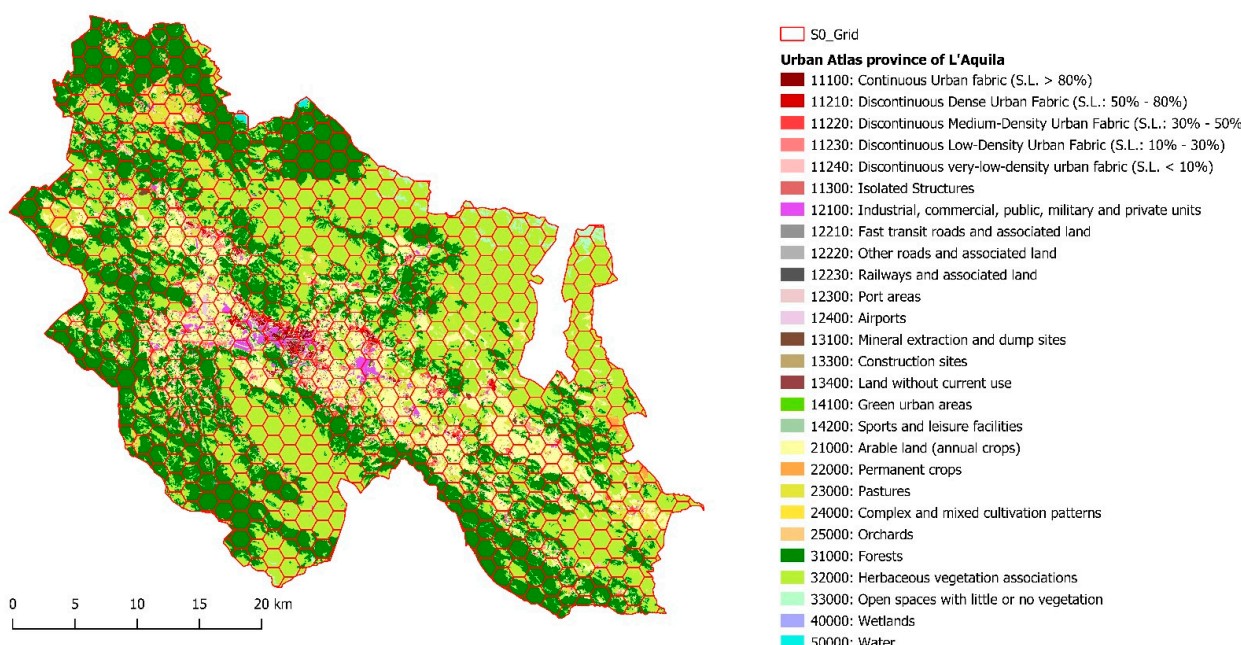

**Figure 4.** First grid intersection ($S_0$—Null shift) with the Urban Atlas dataset.

Clustering algorithms allow analysts to create the desired number of clusters, starting from an input dataset [45]. First, eleven methods were used for this computational task; then, for each method, the "BSS/TSS" metric was calculated. This value represents the ratio of the total between-groups sum of squares (BSS) to the total sum of squares (TSS). Higher values of BSS/TSS entail a better cluster separation [46]. This procedure was executed for each grid shift, extrapolating, among 88 values (one for every clustering algorithm (11) performed for each grid shift (4), first considering only urbanized and non-urbanized areas, then considering all the main categories), the best ones with the belonging method (Table 1).

No data pre-processing was chosen due to the dataset homogeneity.

The criteria set up for the methods were chosen to verify and reduce the possibility of a sensible improvement of the BSS/TSS ratio. The initial re-runs (wherever available) guaranteed an adequate number of reiterations in the choice of the points to be set as cluster centroids. When available, the Manhattan distance was chosen for the calculation of the distances between points. Otherwise, the Euclidean distance was chosen (Table 2). Figure 5 summarizes a comprehensive flowchart of the approach adopted.

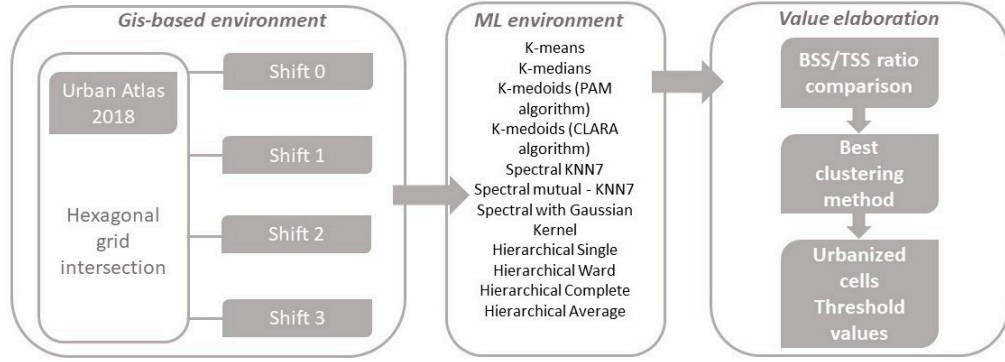

**Figure 5.** Approach flowchart.

**Table 2.** Criteria of the clustering algorithms. The "/" symbol indicates that the setting is absent for the specified algorithm. The ability of the single algorithms to form two different clusters (urbanized areas and not-urbanized areas) from the datasets were evaluated through the ratio of the total between-groups sum of squares (BSS) to the total sum of squares (TSS), or the "BSS/TSS" metric. As the value becomes higher, the cluster separation becomes better.

| METHOD | CRITERIA | | | | | | | |
|---|---|---|---|---|---|---|---|---|
| | Transformation | Initialization Method | Initialization Re-Runs | Max. Iterations | Dist. Function | N. of Samples/ Iterations | Sample Size/Rate | Affinity |
| K—Means | Raw | Random | 150 | 1000 | Euclidean | / | / | / |
| K—Medians | Raw | Random | 150 | 1000 | Manhattan | / | / | / |
| K—Medoids (CLARA Algorithm) | Raw | LAB | / | / | Manhattan | 10 | 200 | / |
| K—Medoids (PAM Algorithm) | Raw | LAB | / | / | Manhattan | / | / | / |
| Spectral K-NN | Raw | Random | 150 | 300 | Manhattan | / | / | K—NN #Neighors = ln(n) + 1 |
| Spectral Mutual K-NN | Raw | Random | 150 | 300 | Manhattan | / | / | Mutual K—NN #Neighors = second ln(n) +1 |
| Spectral Gaussian | Raw | Random | 150 | 300 | Manhattan | / | / | Sigma = $\sqrt{\frac{1}{p}}$ |
| Hierarchical Single linkage | Raw | / | / | / | Manhattan | / | / | / |
| Hierarchical Ward's linkage | Raw | / | / | / | Euclidean | / | / | / |
| Hierarchical Complete linkage | Raw | / | / | / | Manhattan | / | / | / |
| Hierarchical Average linkage | Raw | / | / | / | Manhattan | / | / | / |

## 3. Results

The results are presented in the following section as follows: First, all the results of the BSS/TSS metric for two clustering methods are shown. Then, five analyses of UA classes are compared, focusing on the best one, both for the used number of classes and for the relative clustering algorithm. Finally, the graphical representations for the best clustering methods are deepened, extrapolating the threshold values for the urbanized area of the best one.

### 3.1. BSS/TSS Metric Results

The analyses of two classes show better BSS/TSS ratios for the K—Means method, with the best values of 0.660526 for the first shift ($S_1$) and of 0.659399 for the second one ($S_2$) (Table 3). The variability obtained by shifting the grid with the K—Means method is basically low but increases considering different methods, indicating that the K—Means algorithm performs better in separating the dataset into two clusters. Indeed, the second-best value belonging to the Ward's Linkage method reports a ratio of 0.63072, indicating a worse separation.

**Table 3.** BSS/TSS metric results of the cluster analysis considering, respectively, two and five Urban Atlas classes. K = 2. The better values for each method are underscored.

| METHOD | RATIO OF BETWEEN TO TOTAL SUM OF SQUARES (BSS/TSS) | | | |
|---|---|---|---|---|
| | **TWO UA CLASSES** | | | |
| | $S_0$ | $S_1$ | $S_2$ | $S_3$ |
| K—Means | 0.655676 | 0.660526 | 0.659399 | 0.625655 |
| K—Medians | 0.457219 | 0.466214 | 0.507931 | 0.51609 |
| K—Medoids (PAM and CLARA algorithms) | 0.450213 (CLARA) 0.450213 (PAM) | 0.450298 (CLARA) 0.450298 (PAM) | 0.503916 (CLARA) 0.503916 (PAM) | 0.530062 (CLARA) 0.531839 (PAM) |
| Spectral KNN7 | 0.0431401 | 0.301221 | 0.285217 | 0.369646 |
| Spectral mutual—KNN7 | 0.0172236 | 0.0654304 | 0.025808 | 0.00357514 |
| Spectral with Gaussian Kernel | 0.00069968 | 0.000470756 | 0.103149 | 0.0215007 |
| Hierarchical Single | 0.242457 | 0.170793 | 0.0865453 | 0.37389 |
| Hierarchical Ward | 0.654512 | 0.63072 | 0.656826 | 0.623812 |
| Hierarchical Complete | 0.414637 | 0.564536 | 0.606873 | 0.37389 |
| Hierarchical Average | 0.414637 | 0.564536 | 0.499687 | 0.37389 |
| | **FIVE UA CLASSES** | | | |
| | $S_0$ | $S_1$ | $S_2$ | $S_3$ |
| K—Means | 0.72593 | 0.71977 | 0.7165 | 0.720882 |
| K—Medians | 0.678369 | 0.66402 | 0.670743 | 0.689813 |
| K—Medoids (PAM and CLARA algorithms) | 0.667808 (CLARA) 0.670513 (PAM) | 0.670126 (CLARA) 0.670126 (PAM) | 0.697807 (CLARA) 0.697807 (PAM) | 0.689813 (CLARA) 0.683157 (PAM) |
| Spectral KNN7 | 0.319202 | 0.349396 | 0.17429 | 0.334059 |
| Spectral mutual—KNN7 | 0.0142809 | 0.164442 | 0.116453 | 0.0970793 |
| Spectral with Gaussian Kernel | 0.0175808 | 0.0339564 | 0.0249459 | 0.0296962 |
| Hierarchical Single | 0.0389586 | 0.0268354 | 0.00450728 | 0.0825486 |
| Hierarchical Ward | 0.662854 | 0.646998 | 0.7101 | 0.715121 |
| Hierarchical Complete | 0.542857 | 0.533737 | 0.707919 | 0.622929 |
| Hierarchical Average | 0.0563574 | 0.719427 | 0.700252 | 0.617819 |

The cluster analyses of five UA classes return the best ratios for K—Means, respectively, of 0.72593 for the first grid position ($S_0$) and 0.720882 for the third shift ($S_3$) (Table 3). Regarding the analyses made considering only artificial and non-artificial areas, the variability of the values calculated for the same K—Means method is low but shows substantial differences with the first approach.

The analysis performed using five UA classes shows better values for almost all the methods, but the best ones are still those belonging to the K—Means method. The second

ones, instead of belonging to only one method (such as Ward's linkage for the clustering process performed with two UA classes), belongs to different methods. Referring to the $S_0$ shift, K—Medians has the second-best value, and the $S_1$ shift has this value corresponding to the K—Medoids (PAM and CLARA algorithms) method. The $S_2$ and $S_3$ shifts return Ward's linkage as the second-best clustering method.

### 3.2. Graphic Elaborations and Cell Recognition Ability Comparison

The two UA class analyses show substantial differences, returning values that diverge between the two shifts. The same considerations can be made for the five UA class analyses. It is possible to see how this second approach recognizes a number of urbanized cells returning a close match compared to the UA urbanized axis (Figure 6).

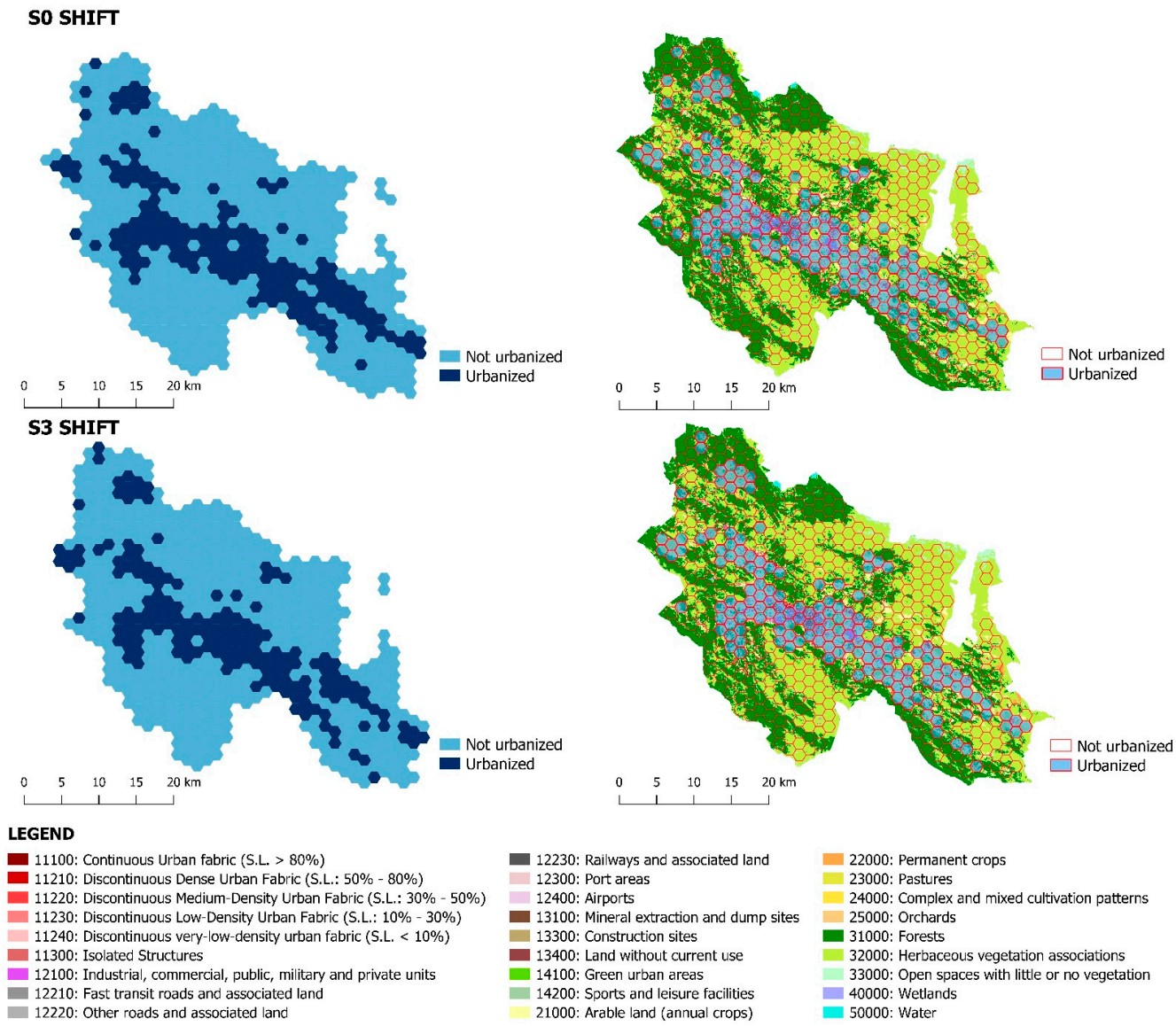

**Figure 6.** K—Means $S_0$ and $S_3$ grid shift classifications using the five main classes from the Urban Atlas dataset.

Furthermore, it can be observed that both the BSS/TSS ratio values of the analysis of the five classes are higher than the ones of the two-class approach, indicating a better cluster separation (Table 3).

Focusing on the urbanized cells classified through the five-classes approach, it is possible to notice that, although the $S_0$ and $S_3$ shifts have the best BSS/TSS values, they

are still not able to recognize some cells. These later become urbanized only partially, as shown in Figure 7, due to the grid translation that leads to a class coverage change.

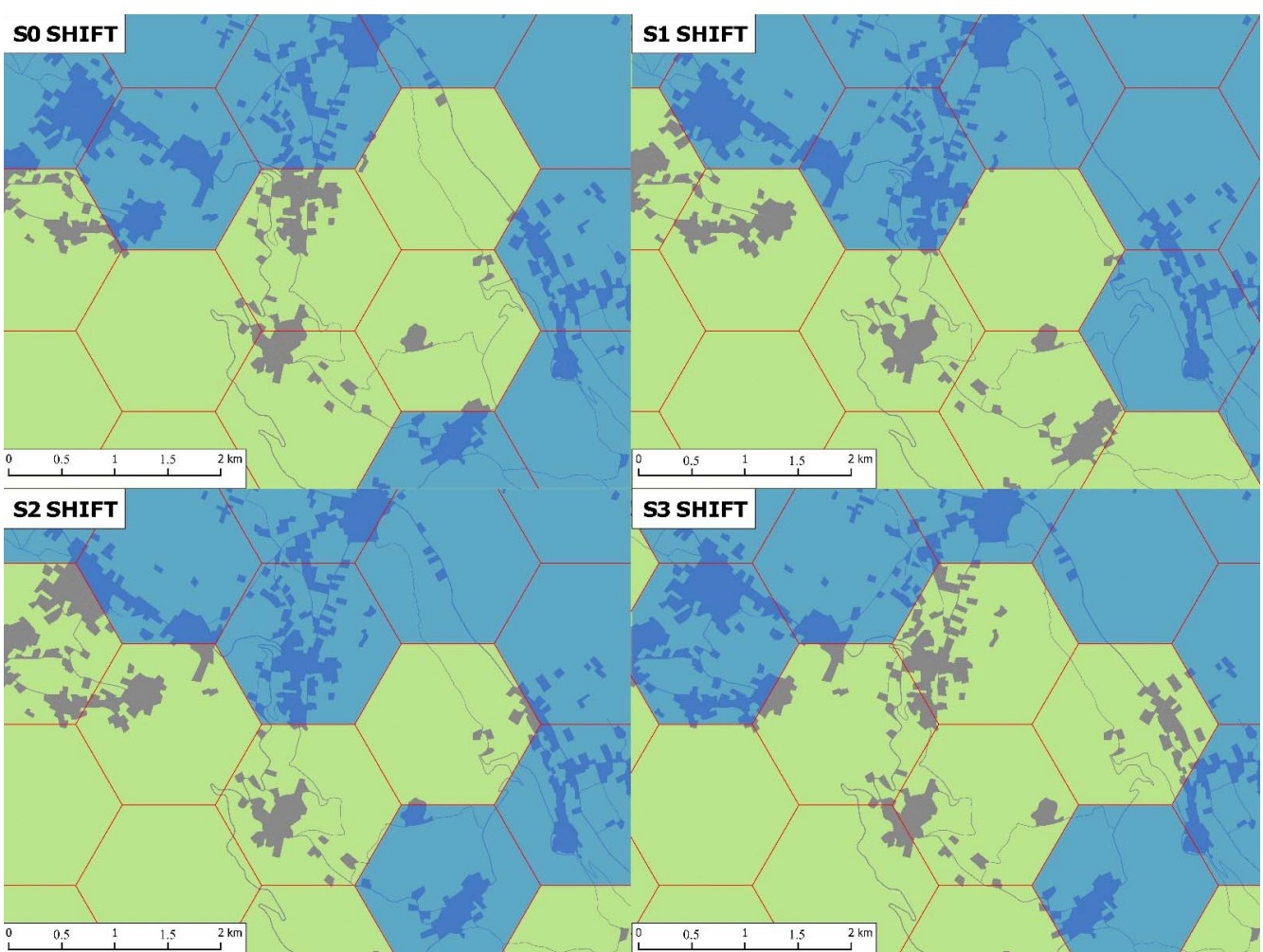

**Figure 7.** Cell recognition ability comparison.

*3.3. Final Elaboration and Implications*

Based on the assumption that the correct definition of the urbanized environment boundaries passes through the best BSS/TSS values of the grid shifts for the best clustering method, the resonance concept here is used to fit the purpose.

Indeed, although five-UA-class K—Means method shows higher BSS/TSS ratio values, none of the single grid translations can return 100% of the urbanized axis.

The shifts can hence be considered as resonance structures. The principle already consolidated in the scientific literature can help the conceptualization of a similar approach in territorial sciences. Every single shift represents a resonance structure, namely one of the possible structures that contributes to defining the real urban boundaries.

In this case, the real conformation of the urbanized axis is identified with the corresponding resonance hybrid, which is deemed the average structure between all the resonance structures, as shown in Figure 8 [47].

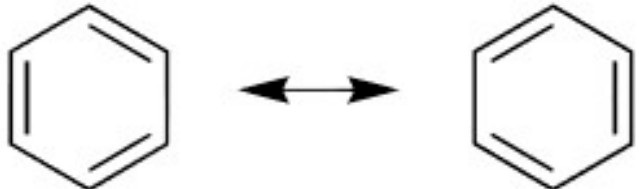

**Figure 8.** Benzene resonance structures [47].

For this, all the urbanized cells from the grid shifts for the five-class K—Means analyses were dissolved, obtaining a probability space that covers all the extension of the urbanized axis, representing the black box which defines the boundaries in which it is possible to perform the desired analyses through hexagonal cell size (Figure 9).

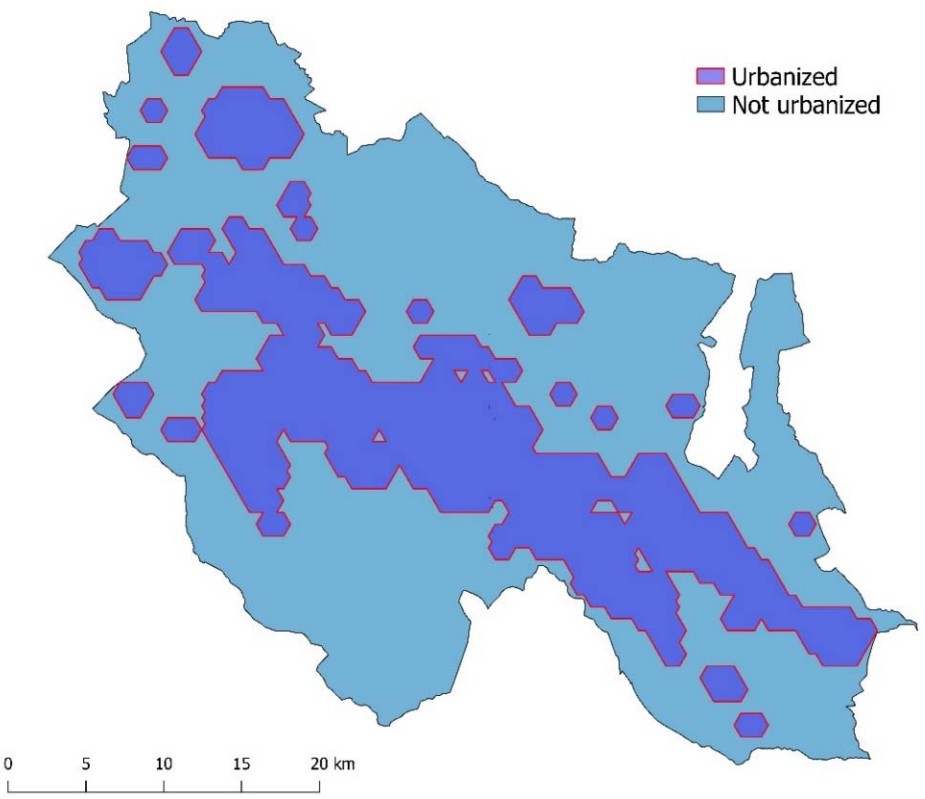

**Figure 9.** Final elaboration obtained dissolving all the urbanized cells resulting from the four K—Means method grid shifts, performed considering five UA classes.

### 3.4. Urban Area Threshold Values

The corresponding values for the urbanized cells, with each of them containing different UA class densities, allowed us to extrapolate the maximum, minimum and mean values relative to the K—Means method (Table 4). The pattern of these cells reveals a mixture of agricultural and artificial areas, accounting on average for 1.4 square km (mean values) of the single cell. Natural areas are the next prevailing class, followed by water bodies (Figure 10).

**Table 4.** Maximum, minimum and mean class values for each grid shift of the best K—Means method (considering only urbanized cells).

| K—MEANS METHOD, FIVE CLASSES OF URBANIZED CELLS | | $S_0$ (km$^2$) | $S_1$ (km$^2$) | $S_2$ (km$^2$) | $S_3$ (km$^2$) | Total Mean (km$^2$) |
|---|---|---|---|---|---|---|
| Artificial areas | Mean | 0.366799 | 0.373849 | 0.381433 | 0.365336 | 0.36606753 |
| | Min | 0 | 0 | 0 | 0.007381 | 0.003690363 |
| | Max | 1.900423 | 1.945933 | 1.915146 | 1.769396 | 1.834909834 |
| Agricultural areas | Mean | 1.024998 | 1.027607 | 1.053506 | 1.064393 | 1.044695669 |
| | Min | 0.05609 | 0.026234 | 0.080825 | 0.094188 | 0.075138679 |
| | Max | 1.936165 | 1.954988 | 1.974852 | 1.975681 | 1.955923058 |
| Natural areas | Mean | 0.604997 | 0.595129 | 0.561606 | 0.567071 | 0.586034235 |
| | Min | 0 | 0.000774 | 0.003879 | 0 | 0 |
| | Max | 1.219838 | 1.230902 | 1.224494 | 1.201435 | 1.210636192 |
| Water bodies | Mean | 0.003124 | 0.003333 | 0.003372 | 0.003115 | 0.003119508 |
| | Min | 0 | 0 | 0 | 0 | 0 |
| | Max | 0.042653 | 0.046287 | 0.046287 | 0.038076 | 0.040364711 |

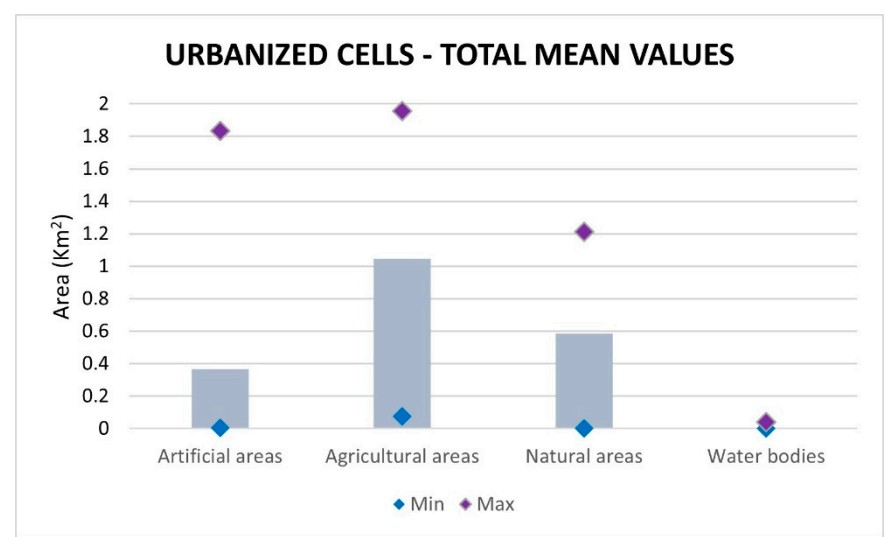

**Figure 10.** Area covered by each UA class, resulting from all the dissolved shifts for the five UA classes K—Means method.

## 4. Discussion

Setting up the grid through the map extension allows for the discrimination of urbanized/non-urbanized cells with a methodology in which the subjectivity of the operator is limited to the choice of cell dimension. Hence, translations can be used to have a standardized shift from which the performances of the clustering methods can be evaluated. This choice improves the ability of single cells to delimit the urban and non-urban environment. Preferring the translation of the cell to random and static positioning ensures that the operator has higher control and efficiency in the analysis of the single tiles [40].

Grid shifting clearly leads to a rearrangement of the frequencies inside the cells, resulting in different performances of the clustering algorithms to recognize them as urbanized/non-urbanized.

Furthermore, the use of the UA dataset poses as a more detailed spatial analysis through which the sector discipline is moving [39]. Being already preprocessed according to uniform standards, this dataset allows one to avoid elaborated preprocessing steps for untrained employees/inexpert users and for a more differentiated result [40].

The choice of the cell dimension of 2 square km supports, contrary to other approaches, the possibility of detecting neighboring portions of the urbanized area, highlighting the

connection with the natural one and, above all, the ecotone continuity between these two systems [48,49].

The cluster analyses performed considering the distinction between urban and non-urban areas return high values of the BSS/TSS ratio for the K—Means method. Despite this, the relative graphic elaborations show a classification that does not manage to reflect the real urbanized axis, as compared to the UA dataset. For this reason, the analyses with the dataset considering the main five Urban Atlas classes were performed. In this case, the results show better clusterization values (BSS/TSS ratio). The highest had an improvement from 0.660526 to 0.72593, also showing closer correspondence with the developed urban axis from the Urban Atlas dataset. This means that, in this case of increasing information in the dataset, due to the use of five different classes instead of the distinction between artificial/non-artificial areas does not lead to a misclassification, but it improves the analyses and the relative graphic representations.

Nevertheless, despite good separation with a BSS/TSS ratio of 0.72593 for the $S_0$ grid shift, some areas still escape the classification method. Indeed, the $S_0$ and $S_3$ shifts, returning the best separation values, are not able to detect some of the cells, which, through the left and down shifts ($S_1$ and $S_2$), become categorized as the urban ones due to a change in their class coverage. The ability of the algorithms to recognize some tiles as urbanized is hence conditioned by the grid position, due to its translations involving a change in the single class coverage inside each cell, leading to a possible different tile classification.

The dissolution of the resulting urbanized cells picked from the best clustering methods for the single grid shift allow for the consideration of all the possible external boundaries of the urbanized area.

The analyses performed for the province of L'Aquila also show a prevalence of agricultural areas. As can be seen in Figure 4, the UA dataset for this territorial scope is characterized by a core artificial area, surrounded by large buffers of agricultural areas. This correspondence is shown clearly in the final elaborations of the current study, returning a prevalence of this latter category for the urbanized cells.

It is also important to consider that the emerged threshold values regarding the urbanized cells only belong to the province of L'Aquila. A different distribution of the artificial areas (subtended by different planning policies and different urban expansion dynamics) can likely lead to variations in the UA class frequencies inside the cells, resulting in variations in the relative threshold values.

Regardless, the latter are a direct consequence of the proposed methodology. By using this, the recurring problem of the definition of threshold values is outdated, leaving the operator with the definition of the quantitative aspects [40].

Another important consequence is the simplicity and uniformity of this approach. Indeed, by using the shifts and the clustering algorithms, the wide variety of data and the task to reorganize them with specific rules are entrusted to the software rather than to the operators themselves, who do not need to understand the heterogeneity of methodologies present in literature, an aspect that can be confusing for those who are approaching these analyses [39].

## 5. Conclusions

The methodology adopted in this article is intended to be functional to the definition of the urbanized tiles of the territorial mosaic compared to the natural ones, independently to the nature of the territory for which this approach is run. The definition of urban boundaries is functional to planning actions and allows for important considerations, such as the study of the dynamics between this system and the surrounding natural areas, as well as the typification of the urbanized mosaic.

The synergy of the territorial sciences with the other scientific sectors of reference is expressed in the role of the control and coordination that these, supported by new technologies, can express the conceptual and technological stitching between urban planning

and land/city regeneration. The areas of research in this direction can be summarized as follows:

- The design of indicators aimed at highlighting the relationships between the sustainable development of urban transformations, the resilience of settlement systems and their potential to adapt to different economic and environmental stresses;
- Machine and deep learning have increasingly been used in urban studies; neural networks were introduced in the 1950s, but only recently they have reached advanced processing power and data storage capabilities to the point that deep learning algorithms can be used to create new applications, including satellite imagery analysis. Therefore, experimentation with ML and DL can only increase the interpretive capabilities of the urban mosaic and generate sustainable city configurations;
- New applications and innovative geospatial services are at the basis of an interdisciplinary development of disciplines, with particular reference to tools for spatial analysis and fast monitoring for urban resilience, the redevelopment of sensitive areas for the contrast of hydraulic and hydrogeological risk, for the mitigation of climate-altering effects, for the improvement of/increase in ecosystem services and for the evaluation of carbon balance.

These applications on the territorial scope represented by the province of L'Aquila show how a ML and DL approach is useful to create standardized methodologies to be shared through easy-to-use software, available both for expert and non-expert users involved in the planning process. This sets a fundamental step toward the comprehension and definition of urban and peri-urban dynamics, addressing the complex task to which territorial policies are called, and introducing other essential planning aspects such as transformation sustainability and emergency/risk management.

It is also clear how, depending on the nature of the settlement dynamics, it is possible to extrapolate threshold values both at a local and large scale, related to the specific territorial scope in which this approach is employed.

Among the limitations of this study, one limitation is the typology of data used. In particular, the Urban Atlas so far is only available for major European cities. This makes the reproducibility of the study limited only to specific areas. Nevertheless, different data sources could be used to overcome this issue.

Another limitation concerns the cell size, which, despite being chosen with certain criteria, may be inadequate to describe different spatial contexts. Therefore, in some cases, it should be necessary to recalibrate this parameter.

Despite the limitations highlighted, the methodology presented in this paper makes it possible to overcome different research gaps related to the discretization of the territory. Indeed, the hexagonal grid and the relative shifts make it possible to overcome the randomized approach of a static grid, which often proves to be ineffective for an efficient recognition process.

This proposed work is part of an experimental framework for characterizing urban settings and assessing relationships with ecosystem services. Future developments are aimed at a deeper application of the method in urban and peri-urban settings.

**Author Contributions:** Conceptualization, L.F.; methodology, A.M., L.F. and F.F.; validation, A.M. and L.S.; formal analysis, L.F. and A.M.; investigation, L.F., A.M., F.F. and L.S.; writing—original draft preparation, L.F. and F.F.; writing—review and editing, A.M. and L.S. All authors have read and agreed to the published version of the manuscript.

**Funding:** This research received no external funding.

**Institutional Review Board Statement:** Not applicable.

**Informed Consent Statement:** Not applicable.

**Data Availability Statement:** For this study we used the Corine Land Cover Data (https://land.copernicus.eu/pan-european/corine-land-cover).

**Acknowledgments:** The methodology was developed within the framework of the Integrated Project LIFE IMAGINE UMBRIA (LIFE19 IPE/IT/000015—Integrated MAnagement and Grant Investments for the N2000 NEtwork in Umbria) and the Project SostEn&Re—sustainability, resilience and adaptation for the protection of ecosystems and physical reconstruction in central Italy.

**Conflicts of Interest:** The authors declare no conflict of interest.

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
