# Peer review of "Discretization of the Urban and Non-Urban Shape: Unsupervised Machine Learning Techniques for Territorial Planning"

_applsci, doi:10.3390/app122010439_

Round 1
Reviewer 1 Report
The article is devoted to the comparison of several methods of cluster analysis. Using cluster analysis, the authors divided the LULC data (aggregated over a regular grid) into two classes. The authors use a hexagonal grid. Using a hexagonal grid is a progressive approach. I support the authors' arguments about the advantages of a hexagonal grid over a rectangular grid. It also seems logical to use grid shift. The authors justify the number of shifts logically and convincingly. The use of a shift is an advantage of the study, which distinguishes it favorably from works where a single grid position is used.
The article can be published after minor edits. It is necessary to clarify what software was used for cluster analysis. You also need to clarify which version of QGIS was used. If any additional plugins for QGIS were used, then you need to specify their name.
There are two typos to correct in the text:
Line 47. An extra parenthesis at the end of a sentence.
Line 165. At the end of the line is written: (Errore. L'origine riferimento non è stata trovata.). Should there be a link to a literary source?
Author Response
Thank you for your kind comments. Please see the attached files for our responses and the revised paper.

Reviewer 2 Report
I had a chance to review article "Discretization of the urban and no-urban shape: unsupervised 2 machine learning techniques and approach to help territorial 3 planning". Indeed the article is an interesting topic and has alot potential to help scientific community and for the readers, however I find some issues in the manuscript which needs to be solved before it can be published.
1. Figures are the biggest concern in this paper as I can hardly read the legends so all figures needs to be updated.
2. Novelty is big concern in this paper as authors failed to identified the research gap which this research will fill.
3. I would have a study area figure just for the readers so that they can know which area we are talking about.
4. Very little explanation is given for the models used in this study, it needs more explanation.
5. I find discussion is similar to results because authors failed to connect their finding with other published research and how their results are better or otherwise.
6. What are the assumptions, implications and limitations of this research? Kindly mention.
Author Response

(The authors gave the same response as above.)

Round 2
Reviewer 2 Report
I think the authors made sound progress and paper can be published now.